# Nasolabial Appearance in 5-Year-Old Patients with Repaired Complete Unilateral Cleft Lip and Palate: A Comparison of Two Different Techniques of Lip Repair

**DOI:** 10.3390/jcm11102943

**Published:** 2022-05-23

**Authors:** Sonja Lux, Matthias Mayr, Michael Schwaiger, Sarah-Jayne Edmondson, Christoph Steiner, Peter Schachner, Alexander Gaggl

**Affiliations:** 1Department of Oral and Maxillofacial Surgery, University Clinic Salzburg, 5020 Salzburg, Austria; s.lux@salk.at (S.L.); m.mayr@salk.at (M.M.); c.steiner@salk.at (C.S.); p.schachner@salk.at (P.S.); a.gaggl@salk.at (A.G.); 2Department of Oral and Maxillofacial Surgery, Medical University of Graz, 8036 Graz, Austria; 3South Thames Cleft Service, Guy’s and St. Thomas’ Hospital, London SE1 9RT, UK; edmondsonsj@gmail.com

**Keywords:** cleft lip, surgical procedures, aesthetics, maxillofacial abnormalities, congenital defects

## Abstract

Different surgical techniques are available to adequately correct the primary cleft lip deformity; however, when compared, none of these techniques have proven superior with regard to achieving optimal aesthetic results. Thus, the aim of this retrospective study was to assess the nasolabial appearance in patients with unilateral cleft lip and palate (UCLP) at age five with reference to two techniques for primary cleft lip repair used in our service: Pfeifer’s wave-line procedure and Randall’s technique. A modified Asher–McDade Aesthetic Index was applied to appraise the nasolabial area by means of 2D photographs of non-syndromic five-year-old patients with a UCLP. In this context, three parameters were assessed: 1. nasal frontal view; 2. shape of the vermilion border and philtrum length; and 3. the nasolabial profile. Five professionals experienced in cleft care were asked to rate the photographs on two occasions. Overall, 53 patients were included in the final analysis, 28 of whom underwent lip repair according to Pfeifer; 25 were treated employing Randall’s technique. Statistically significant differences between the two techniques regarding philtrum length and vermilion border were found (*p* = 0.046). With reference to the other parameters assessed, no significant differences were determined. The results suggest that Randall’s cleft lip repair may allow for more accurate alignment of the vermilion border and more adequate correction of the cleft lip length discrepancy in comparison to Pfeifer’s wave-line technique.

## 1. Introduction

‘Cleft lip and palate’ (CLP) is one of the most common congenital deformities, occurring with an incidence of 1 in 700 live births in Caucasian populations [1]. What is more, an increasing prevalence of CLP has been noted in recent years, which may, among other reasons, be related to a reduced neonatal mortality rate, more accurate documentation of CLP, and the influence of numerous extrinsic factors triggering CLP [2,3]. Several contributing factors have been identified that are closely linked to the occurrence of CLP, namely genetic and environmental parameters. In terms of genetics, CLP has commonly been linked to the genetic loci IRF6, VAX1, and PAX7 [4]. With regard to environmental factors, smoking, maternal age, BMI, and gestational diabetes were most closely associated with the occurrence of CLP [5].

The treatment of CLP involves elaborate and complex surgical intervention, its extent depending on the type and severity of the cleft. Great advances in cleft care have been made over the last few decades; however, children born with CLP continue to have increased morbidity [6]. Particularly in this context, the significance of facial aesthetics in patients with unilateral and bilateral cleft lip and/or palate (UCLP or BCLP, respectively) is frequently highlighted throughout the literature, demonstrating a correlation between a cleft patient’s facial appearance and social stigmatization [7].

The goal of cleft surgery is multifactorial but in order to prevent the aforementioned issues and to minimize the psychological impact of CLP, surgeons aim to recreate the natural-appearing anatomy of the lip and nose during primary surgery. Numerous surgical techniques have been described for the correction of a unilateral cleft lip deformity, ranging from: straight line [8] and “wave-line” incisions that result in a straight scar in the region of the philtrum [9]; “rotation advancement” flaps [10,11]; the “triangular” flap [12,13]; to a hybrid technique called the “anatomic subunit approximation” concept of lip repair (the “Fisher” repair) [14] and various modifications of this. A variety of studies have investigated aesthetic outcomes related to different surgical techniques for primary cleft lip repair, most of which have relied on two-dimensional (2D) photographic assessment [15,16,17]. At present, no consensus has been reached as to which technique yields the best aesthetic results.

There are a variety of established assessment tools that can be used to evaluate the nasolabial appearance in cleft patients in both two- and three-dimensional (2D and 3D) formats [18]. Amongst those assessment tools, the Asher–McDade Index is one that is widely used and specifically focuses on the assessment of the lip and nose in cleft patients on the basis of two-dimensional photographs [19].

In our institution, two different surgical techniques for primary cleft lip repair are used as standard in the correction of cleft lip deformity: Pfeifer’s “wave-line” procedure and Randall’s “triangular flap” technique [9,13]. Given the scarcity of data on this topic, the aim of our study was to assess and compare the nasolabial appearance in patients with UCLP at age five to determine with of the aforementioned surgical techniques for primary lip repair results in a more favorable aesthetic outcome.

## 2. Materials and Methods

### 2.1. Study Design

Patients with a complete UCLP that underwent primary cleft lip repair and subsequent palatoplasty in our department between 1995 and 2005 were eligible for inclusion in this retrospective study. The patients also had to fulfil the following inclusion criteria: (1) completed 5-year audit; (2) non-syndromic status; (3) surgical technique: primary lip repair of either Pfeifer’s “wave-line procedure” or Randall’s “triangular flap” technique [9,13]; (4) primary lip and palate repair only, with no further corrective surgeries before the age of five; (5) procedure performed by a consultant cleft surgeon; (6) standardized 2D photographs taken in frontal and lateral profile view as part of the five-year follow up (distance 1 m; a swivel chair and floor markers for 0 and 90 degrees); and (7) no orthodontic treatment, except passive presurgical orthopedics (Hotz-plate) [20].

### 2.2. Surgical Technique

In Pfeifer’s approach [9] and Randall’s technique [13], similar anatomical landmarks are taken into account to address the underlying deformity. This specifically refers to the lip markings at the philtrum border on the non-cleft side and the medial aspect of the cleft, as well as the markings at the vermilion border on the lateral side of the cleft.

#### 2.2.1. Pfeifer’s “Wave-Line Technique”

The following surgical principles are deployed with regard to Pfeifer’s approach [9]: medially, an incision in the form of a half or full wave is designed; on the lateral segment, an arched-shaped incision is performed. Stretching the aforementioned incisions should then allow for adequate correction of the lip length discrepancy. The design of the wave on the medial segment varies according to the amount of the lip length discrepancy noted. To facilitate the accurate alignment of the cleft lip, a bendable wire matching the length of the non-cleft side is used to design the wave on the medial and lateral side of the cleft lip. The lip scar resulting from Pfeifer’s approach is a straight line located at the philtrum border.

The lip markings of Pfeifer’s “wave-line” technique are illustrated in Figure 1.

#### 2.2.2. Randall’s “Triangular Flap” Technique

In Randall’s “triangular flap” technique, in marked contrast to Pfeifer’s technique, the lip length discrepancy is addressed very differently by designing a triangular flap above the white roll on the lateral side of the cleft lip that will sit into a back-cut incision at the medial labial segment [13]. The size of the triangular flap and the length of the back-cut correspond to the lip length discrepancy determined between the non-cleft side and the medial side of the cleft lip. The angulation of the triangular flap and the back-cut made may vary depending on the width of the cleft and the tissue available to repair the lip. The lip scar consists of a straight line in combination with a small triangle above the white roll.

The lip markings of Randall’s “triangular flap” technique are shown in Figure 2.

#### 2.2.3. Surgeons

All operations were performed by two consultant cleft surgeons.

### 2.3. Aesthetic Outcome Assessment

For subjective assessment of the nasolabial appearance, a modification of the Asher–McDade Index as described by Brusati [21] was used (Figure 3).

Before the assessment, 2D photographs were cropped according to the Cleft Aesthetic Rating Scale (CARS) [22] (Figure 4). A five-point Likert scale was used to assess each of the aforementioned parameters: (1) very good appearance; (2) good appearance; (3) fair appearance; (4) poor appearance; and (5) very poor appearance (Figure 5).

Five professionals experienced in cleft care (three oral and maxillofacial surgeons, one speech and language therapist, and one orthodontist), employed in our service scored the cropped photographs. The assessors were blinded to the patients’ identities. The two surgeons who had performed the procedures were not among the raters.

#### 2.3.1. Nasal Frontal View (NF)

Within this modification of the Asher–McDade Index, the two categories ‘nasal form’ and ‘deviation/symmetry of the nose’ in the frontal view were combined and were referred to as ‘nasal frontal view’ (NF).

#### 2.3.2. Vermilion Border/Philtrum Length (VB/P)

The second parameter evaluated in the frontal view was the shape of the vermilion border. In addition, the symmetry and length of the philtrum on the cleft side were assessed and compared to the non-cleft side as described by Mosmuller et al. [22]. This parameter was named vermilion border/philtrum length (VB/P).

#### 2.3.3. Nasolabial Profile (NLP)

In the lateral view, the third parameter specifically focusing on the sagittal nasolabial profile was assessed (NLP). The latter parameter only took into account the cleft side and photographs of patients with a left-sided cleft were mirrored so that all photographs appeared to be taken from a right lateral view.

Raters were allowed to browse through the slides prior to rating them to become familiar with the types of images to be evaluated. The same assessors were asked to score the cropped photographs three months later in a different randomization order.

### 2.4. Statistical Analysis

Statistical analysis was performed using IBM SPSS Version 24.0 (Windows 7, IBM, Armonk, NY, USA). Unpaired T-tests were used to compare the results of the two groups after the normal distribution was tested using the Kolmogorov–Smirnov Test. Intraclass correlation was assessed by means of the one-way random single measure reliability analysis, which was calculated for each observer (ICC 1.1). Interclass correlation was assessed using the two-way mixed ICC (ICC 3.1) at the first evaluation time point (T1).

## 3. Results

### 3.1. Demographics

Overall, 53 patients with a complete UCLP were included in the final analysis. Of these, 28 patients underwent lip repair according to Pfeifer’s wave-line procedure (group ‘Pfeifer’), and 25 patients had their lips repaired according to Randall’s technique (group ‘Randall’). In both groups, lip repair was aimed to be performed at an age of 6 months (average 6.4 months (±1.4) ‘Pfeifer’ and 6 months (±1.4) ‘Randall’); subsequent palatoplasty was performed at an age of 12 months. None of the patients underwent gingivoperioplasty for alveolar closure. Further demographics are shown in Table 1.

### 3.2. Inter-Rater and Intra-Rater Reliability

The rating mechanism was first tested for reliability using the inter-class correlation (ICC). The ICC was 0.681 (95% confidence interval: 0.573; 0.778), indicating fair agreement between the assessors. The intra-class correlation coefficient was calculated for every rater individually. The values ranged from 0.837 (95% confidence interval: 0.710; 0.907) to 0.656 (95% confidence interval: 0.046; 0.860). The average intra-class correlation amounted to 0.753, suggesting good agreement for each rater.

### 3.3. Aesthetic Score

#### 3.3.1. Vermilion Border and Philtrum Length (VB/P)

The average ratings for vermilion border and philtrum length (VB/P) were 2.51 (±0.92) and 2.03 (±0.77) in the ‘Pfeifer’ and ‘Randall’ groups, respectively, resulting in a statistically significant difference between the two (*p* = 0.046).

The patient with the worst subjective outcomes in the ‘Pfeifer’ group had an average rating of 4.3; six patients had average ratings of less than 3.5. In the ‘Randall’ group, the lowest average rating in a single patient amounted to 3.3.

#### 3.3.2. Nasal Frontal View (NF)

The average rating for the nasal frontal (NF) view was 2.64 (±0.71) in ‘Pfeifer’ and 2.42 (±0.57) in ‘Randall’ groups, respectively. The lowest rating in ‘Pfeifer’ was 4.0 (three patients had a rating lower than 3.5). In the ‘Randall’ group, the lowest rating was 3.2. No statistically significant differences were found. 

#### 3.3.3. Nasolabial Profile (NLP)

The overall rating of the nasolabial profile (NLP) was 2.50 (±0.67) in ‘Pfeifer’ and 2.54 (±0.83) in ‘Randall’ groups, respectively. For NLP, the lowest rating in ‘Pfeifer’ was 4.2 and in ‘Randall’ was 4.4. Here, one patient in ‘Pfeifer’ and three in ‘Randall’ had ratings lower than 3.5. No statistically significant differences between the techniques were found. 

All parameters and respective statistical results are shown in Table 2.

## 4. Discussion

Patients with CLP and the resultant difference in facial appearance are commonly referred to as ‘social’ pathologies [23]. This is because a patient’s facial appearance, expressions, mimicry, and speech may be severely affected by the underlying deformity, all of which play an essential role within social interaction and perception. In this context, it has been shown that facial anomalies are often subject to poor social acceptance and, as a result of this, children with CLP frequently experience stigmatization and taunting [23]. Social acceptance is considered extremely important in acquiring positive self-perception and self-esteem, especially in periods of a changing social environment. Such a change usually occurs around the age of five when entering school, where children with CLP may first be confronted with being different from their peers [23,24].

Against this backdrop, it comes as no surprise that aesthetic and functional outcomes tend to have an enormous impact on patients with CLP and their parents [25]. To minimize the negative sequelae related to orofacial clefting with regard to a patient’s social integration, cleft surgeons continue to strive for optimal treatment strategies, aiming to achieve the best possible surgical results. With reference to primary cleft lip repair in complete unilateral cases, the latter includes optimal muscle alignment allowing for ideal lip function, adequate lip length, symmetrical proportions, and a well-concealed, hardly visible scar.

As previously mentioned, numerous surgical techniques have been established with the aid of which the unilateral cleft lip and nasal deformity can adequately be addressed [8,9,10,11,12,13,14]. Authors have aimed to compare techniques by means of subjectively and objectively reviewing outcomes [15,16,17,26,27]. However, while interesting findings have been revealed, none of the techniques available have proven superior with regard to reliably achieving optimal aesthetic outcomes. Given the lack of consensus and the great importance pertaining to this specific topic, the present study focused on subjectively evaluating aesthetic outcomes related to two different primary cleft lip repair techniques, which have not yet been compared in this regard (‘Pfeifer’ vs. ‘Randall’). At this stage, it needs to be highlighted that, besides the technique used, numerous other co-founding parameters, such as a surgeon’s skills and preferences, the initial cleft lip severity and the timing of the lip repair may significantly affect aesthetic surgical outcomes. In light of these contributing factors potentially affecting outcomes, we aimed to reduce potential bias to a minimum by ensuring a homogenous study cohort in terms of the timing of surgery, the type and severity of the underlying deformity, the surgeons’ sets of skills, and experience, as well as ensuring the perioperative management and follow-up treatment were uniform. In addition, no further corrective surgical procedures to the lip and nose were performed after primary surgery and all patients assessed were of a similar age (five years). This specific age group was selected due to the great psychosocial impact triggered by a patient’s facial appearance around that age [23].

Standardized two-dimensional photographs were chosen to assess the patient cohort. The assessment tool used for this purpose was the well-validated Asher–McDade index and minor modifications of this index, as described above [19,21,22]. These modifications were considered to facilitate the rating and hence were chosen over the original description of the index.

Based on the ratings of the five assessors who were asked to subjectively analyze the photographs in a randomized order, the following results were determined: with regard to nasal proportions and symmetry in the nasal frontal view (NF), no statistically significant differences amongst the techniques compared were observed. Similarly, there were no significant differences in terms of the nasolabial profile (NLP). The mean ratings concerning these categories ranged between the levels 2 and 3, which are good/fair overall results. This indicates that, from a subjective point of view, both of these techniques were linked to acceptable nasal proportions and a nasolabial profile, with no relevant differences detected.

Conversely, significant differences regarding the vermilion border and philtrum length were found to occur (VB/P) in our cohort (*p* = 0.046), as group ‘Pfeifer’ was scored lower compared to group ‘Randall’. It is acknowledged, though, that there was only a fairly significant difference determined, and that, therefore, the statistical findings do only allow to draw tentative conclusions. Still, when reviewing these results, it is suggested that the worse ratings in group ‘Pfeifer’ may be linked to a shortening of the upper lip on the cleft side. This has previously been quoted by Grundlach et al. [28]. Furthermore, this was also noticed by the senior cleft surgeon in our service, and while his observations were purely subjective, he went on to prefer Randall’s technique over that introduced by Pfeifer later in his career and after the time period assessed in this study [9,13]. In an attempt to quantify these perceptions, the present study was conducted. The assessors were kept unaware of the personal views of the senior cleft surgeon regarding the advantages and shortcomings of the techniques applied, so as not to bias the ratings.

Another issue that should be addressed before contemplating our findings in more detail is the surgeons’ skill, which very likely improved throughout their careers. We accept that this may be a confounding factor affecting our results, while stressing the fact that both surgeons had already acquired advanced cleft surgical skills at the beginning of the period analyzed.

What is more, we appreciate that the results obtained were based on subjective ratings, rather than on objective measurements. In this regard, it has previously been suggested that an examiner’s professional background and familiarity with the topic may significantly affect the ratings. In a review on subjective outcomes related to cleft lip and palate surgery, laypersons were regularly found to be more critical [29]. This may be related to the fact that, while professionals tend to put outcomes into perspective in terms of the complexity of the surgery and the severity of the underlying deformity, laypersons may be less biased in this context. Having said that, some authors suggested the exact opposite, as professionals were shown to rate more critically than laypersons [29]. With reference to the present work, all assessors can be considered cleft care professionals, two of whom had a non-surgical background. The intra-rater reliability amounted to 0.753, which shows good agreement. This implies, that the examiners were consistent with their ratings and did not change their minds between the first and second assessments. In terms of the interrater-reliability, which is the more interesting parameter to look at in this context, moderate agreement amongst the raters was found (0.681); however, no distinct differences between surgical and non-surgical ratings were detected. Whilst we concede that the non-surgical group of raters was unaware of the peculiarities regarding the techniques for the lip repair applied, it is appreciated that they still had a certain level of expertise with regard to aesthetic surgical outcomes related to primary cleft lip repair. For future studies, it would be interesting to additionally include laypersons to further investigate the effect of the social and professional background on the subjective perception of facial aesthetics in cleft surgery. This may prove particularly useful, as a child’s awareness of being different from its peers mainly originates from its peers, rather than from professional caretakers [23].

An issue that has recently been contemplated when it comes to analyzing aesthetic outcomes in cleft lip and palate surgery is that most studies did not take into account PROMs (patient-reported outcome measures) [30]. It seems plausible that a patient’s perception might be entirely different from an ‘objective’ rater’s view, and hence, the use of PROMs may have helped to better understand the psychosocial impact of the cleft. This, however, was considered problematic to be included in our study owing to its retrospective character. Whilst we acknowledge that this would have been an interesting addition to look at in our patients, we decided not to include PROMs and to accept the limitations arising in this context. To better objectify outcomes related to cleft lip surgery, anthropometric measurements have been proposed [26,27]. A good example of how to adequately measure nasolabial proportions is Patel’s study, which compared two surgical techniques for primary cleft lip repair based on a two-dimensional anthropometric assessment. Numerous parameters analyzed in this regard corresponded to those assessed on a subjective level with regard to the Asher–McDade index [19]. Thus, a combination of both subjective ratings and objective measurements appears reasonable, which has previously been advocated for by other studies in this field [30]. In this connection, we suggest that 3D analysis may be more accurate in terms of objective anthropometric measurements, and whilst 3D photographic assessment was introduced in our service a few years ago, this tool was not available for every patient included in the present study. What is more, this paper’s aim was directed towards identifying differences between Randall’s and Pfeifer’s techniques based on subjective perception, and thus objective measurements were not included. The benefit of 3D photographs in the context of subjective photographic assessment is questionable according to our experience.

## 5. Conclusions

Both surgical techniques assessed were rated to achieve good overall results at the age of five in terms of the nasolabial appearance. It is suggested that Randall’s technique for primary cleft lip repair may allow for a more accurate alignment of the vermilion border and more adequate correction of the cleft lip length discrepancy in comparison to Pfeifer’s wave-line technique while highlighting again that only a fairly significant difference was observed when comparing the two techniques in terms of the parameter VB/P (*p* = 0.046). To further minimize the risk of potential bias and quantify our findings, a prospective randomized controlled clinical trial would have to be conducted, randomizing the order of the techniques applied. With regard to Pfeifer’s approach, it is proposed that, while a subtle lip scar can be achieved when aligned correctly, its wave-like design seems more susceptible to planning error in comparison to Randall’s technique. The latter technique, in turn, is based on distinct geometrical principles to address the cleft lip deformity, which, according to our experience, appear to facilitate accurate alignment of the vermilion border and correction of the lip length discrepancy. Regarding our center, this led to Pfeifer’s technique being abandoned for primary cleft lip repair in complete unilateral cases and to Randall’s technique being entrenched as the method of choice in this context.

It needs to be acknowledged, though, that nasolabial proportions and symmetry may still be subject to change during growth. What is more, even the best surgical results may continue to be linked to stigmatization. Hence, we propose that prospective studies are required to re-evaluate and quantify our findings regarding different age groups, while additionally considering the psychosocial impact of the cleft in more detail.

## Figures and Tables

**Figure 1 jcm-11-02943-f001:**
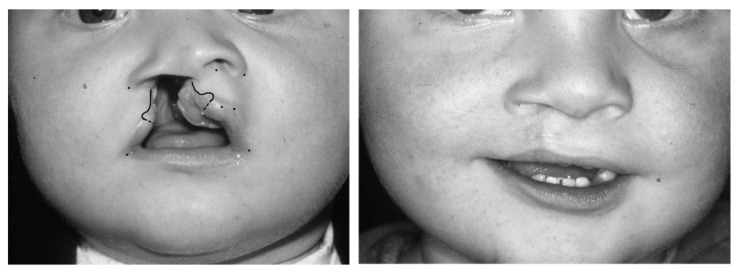
Indicating the lip markings according to Pfeifer’s technique [9].

**Figure 2 jcm-11-02943-f002:**
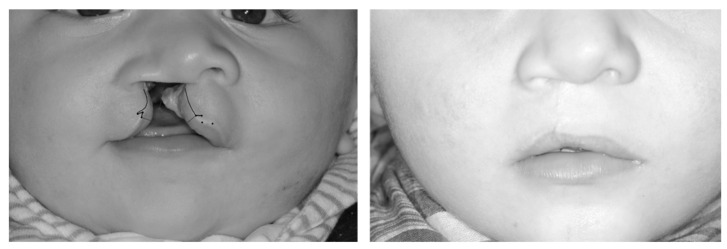
To show the lip markings with reference to Randall’s technique [13].

**Figure 3 jcm-11-02943-f003:**
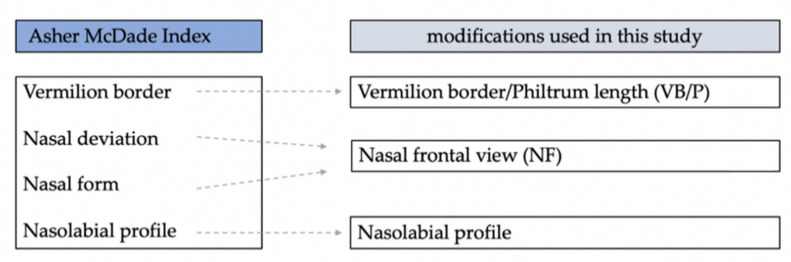
To show the parameters assessed within the original description of the Asher–McDade Index and the modifications used in this study [19,21,22].

**Figure 4 jcm-11-02943-f004:**
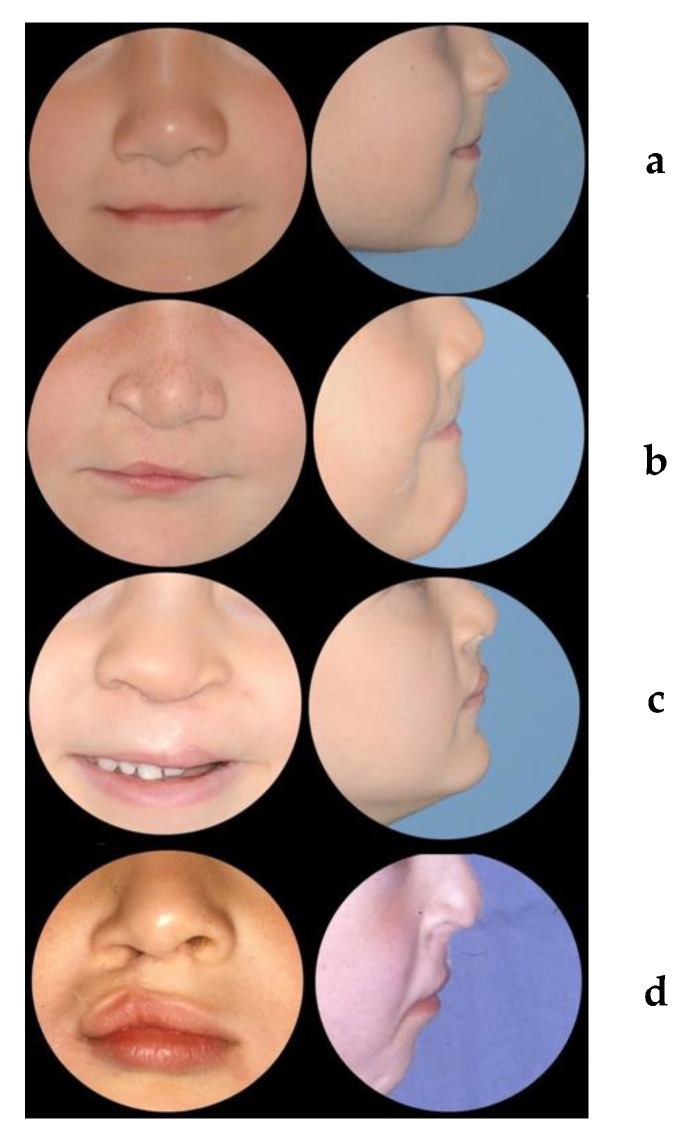
To show the cropped photographs of patients whose nasolabial appearance was unanimously perceived as (**a**) very good; (**b**) good (**c**) fair; and (**d**) poor, according to the scoring system used [19,21,22].

**Figure 5 jcm-11-02943-f005:**
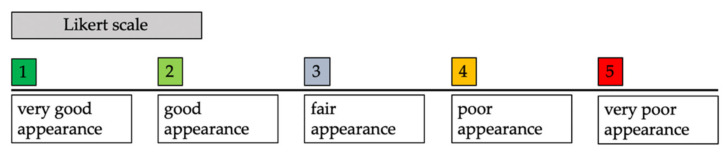
To display the Likert scale used for assessment of the cropped photographs.

**Table 1 jcm-11-02943-t001:** To show the demographics of the study cohort. (m: male; f: female; r: right; l: left).

	‘Pfeifer’ Group	‘Randall’ Group
Number	28	25
Sex	m: 16 f: 12	m: 14 f: 14
Average age at lip repair (months)	6.4 (±1.4)	6.0 (±1.4)
Minimum/maximum age at lip repair (months)	4.7–7.9	4.3–7.7
Average age at photograph (months)	60.6 (±1.9)	60.8 (±2.0)
Cleft side	r: 12; l: 16	r: 13; l: 12

**Table 2 jcm-11-02943-t002:** Displaying the statistical results determined according to respective parameters NF (nasal frontal); VB/P (vermilion border and philtrum length) and NLP (nasolabial profile); (SD—standard deviation; CI—confidence interval; mean diff.—mean difference).

Parameters	‘Pfeifer’ Group					‘Randall’ Group					Mean Diff.	*p*-Values
	Min	Max	Mean	SD	95% CI	Min	Max	Mean	SD	95% CI		
**NF**	1.20	4.00	2.64	0.71	[2.30–2.88]	1.00	3.20	2.42	0.57	[2.18–2.65]	0.22	*p* = 0.219
**VB/P**	1.00	4.30	2.51	0.92	[2.10–2.87]	1.00	4.00	2.03	0.77	[1.71–2.35]	0.48	*p = 0.046*
**NLP**	1.30	4.20	2.50	0.67	[2.23–2.77]	1.30	4.40	2.54	0.83	[2.19–2.88]	−0.03	*p* = 0.880

## Data Availability

Not applicable.

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
