# Peer review of "Nasolabial Appearance in 5-Year-Old Patients with Repaired Complete Unilateral Cleft Lip and Palate: A Comparison of Two Different Techniques of Lip Repair"

_jcm, 2022, doi:10.3390/jcm11102943_

Round 1

Reviewer 1 Report

This article has been well written and described. It reviewed UCLP patients underwent two surgical techniques between 1995 to 2005. Asher McDade Index were used for nasolabial appearance. The results revealed that Randall’s technique had more accurate alignment of the vermilion border  comparison to Pfeifer’s wave line-technique. The nasoappearance has no difference. 

the comments are as follows:

  1. It would be better to provide a longer follow up for those patients since those patients were underwent surgical correction between 1995 to 2005.
  2. Did the patients receive primary rhinoplasty? if not, then the nasofrontal view and the nasolabial profile were expected to no difference between the group. The study only told us the Pfeifer’s wave line-technique has the problem of aling the vermillion border. 

Author Response

Comment 1

This article has been well written and described. It reviewed UCLP patients underwent two surgical techniques between 1995 to 2005. Asher McDade Index were used for nasolabial appearance. The results revealed that Randall’s technique had more accurate alignment of the vermilion border comparison to Pfeifer’s wave line-technique. The naso-appearance has no difference. 

the comments are as follows:

It would be better to provide a longer follow up for those patients since those patients were underwent surgical correction between 1995 to 2005.

Authors’ reply:

The reviewer is right, that it would be beneficial to analyze this patient cohort in terms of its nasolabial appearance taking into account a longer follow up period.

However, this was not done for the following reasons:

  • to ensure that no further corrective procedures of the lip and nose had been performed before the photographic assessment.
  • to ensure very standardised time-points with regard to the photographs taken.
  • to ensure that none of the patients included had fixed orthodontic appliances, because this would have affected the nasolabial appearance (lip projection, nasolabial angle etc.).

Comment 2

Did the patients receive primary rhinoplasty? if not, then the nasofrontal view and the nasolabial profile were expected to no difference between the group. The study only told us the Pfeifer’s wave line-technique has the problem of aling the vermillion border. 

Authors’ reply:

In none of the patients included primary rhinoplasty was performed (apart from trying to adequately align the alar base and define the width of the nostrils). While we agree that no differences with regard to the torque of the LLCs and the position of the nasal septum was to be expected, we suggest, however, that different techniques for cleft lip repair do influence how well the alar base can be aligned, how accurate the width of the nostrils can be defined and how the nasal entrance can be shaped. We appreciate, though, that from a subjective point of view no statistically significant differences were determined with regard to the nasal appearance when comparing the two techniques in the present study. What is more, we would like to stress that shortening of the lip on the cleft side and insufficient alignment of the vermilion border may significantly affect a patient’s facial appearance and facial harmony. Thus, we propose that the findings determined in our study may add crucial information to a cleft surgeon’s daily practice and decision making in terms of cleft lip repair.

Reviewer 2 Report

Journal: Journal of Clinical Medicine

Manuscript ID: jcm-1687450

Type of manuscript: Research

Title: Nasolabial Appearance in 5-Year-Old Patients with Repaired Complete Unilateral Cleft Lip and Palate: A Comparison of Two Different Techniques of Lip Repair

This manuscript was intended to assess and compare the nasolabial area of five-year-old patients with a UCLP who underwent cleft lip repair using two different surgical techniques in aim to identify the surgical technique with a more favorable aesthetic outcome.

This article with diligent methodology of research is correctly prepared and well-written. The only problem is that it leaves the reader almost at the same point as at the beginning of the article’s introduction –  “A variety of studies have investigated aesthetic outcomes related to different surgical techniques for primary cleft lip repair, most of which have relied on 2-dimensional (2D) photographic assessment [15-17]. At present, no consensus has been reached as to which technique yields the best aesthetic results.” What’s worse, the novelty can not be seen in the Asher McDade Index which is widely used in the assessment of the lip and nose in cleft patients on the basis of 2-dimensional photographs either. Additionally, surprisingly the small numbers of compared groups (the cleft lip repair is one of the most frequently performed procedures in cleft surgery) and the age of patients at evaluation (at the beginning of face growth changes) suggest that the authors were very determined to publish an article in the field of cleft surgery.

Unfortunately, it seems that the authors have not made many steps forward by suggesting only that Randall’s technique (probably the most popular one allover the world) for primary cleft lip repair allows for a more accurate alignment of the vermilion border and more adequate correction of the cleft lip length discrepancy in comparison to Pfeifer’s wave line-technique.

Coming up to the end, at least I would like to know if the conclusions of this study persuaded the surgeons of the authors’ center to discontinue the Pfeifer’s wave line-technique in favor of Randall’s technique for primary cleft lip repair - and this should be clearly stated at the end of the article.

Author Response

REVIEWER 2

This manuscript was intended to assess and compare the nasolabial area of five-year-old patients with a UCLP who underwent cleft lip repair using two different surgical techniques in aim to identify the surgical technique with a more favorable aesthetic outcome.

This article with diligent methodology of research is correctly prepared and well-written. The only problem is that it leaves the reader almost at the same point as at the beginning of the article’s introduction –  “A variety of studies have investigated aesthetic outcomes related to different surgical techniques for primary cleft lip repair, most of which have relied on 2-dimensional (2D) photographic assessment [15-17]. At present, no consensus has been reached as to which technique yields the best aesthetic results.” What’s worse, the novelty cannot be seen in the Asher McDade Index which is widely used in the assessment of the lip and nose in cleft patients on the basis of 2-dimensional photographs either. Additionally, surprisingly the small numbers of compared groups (the cleft lip repair is one of the most frequently performed procedures in cleft surgery) and the age of patients at evaluation (at the beginning of face growth changes) suggest that the authors were very determined to publish an article in the field of cleft surgery.

Unfortunately, it seems that the authors have not made many steps forward by suggesting only that Randall’s technique (probably the most popular one all over the world) for primary cleft lip repair allows for a more accurate alignment of the vermilion border and more adequate correction of the cleft lip length discrepancy in comparison to Pfeifer’s wave line-technique.

Coming up to the end, at least I would like to know if the conclusions of this study persuaded the surgeons of the authors’ center to discontinue the Pfeifer’s wave line-technique in favor of Randall’s technique for primary cleft lip repair - and this should be clearly stated at the end of the article.

Authors’ reply:

We would like to thank the reviewer for reviewing our manuscript and for highlighting specific shortcomings with regard our paper. The reviewer is correct that conclusions drawn from this study need to be stated more clearly, and, hence, these were added to the manuscript.

What is more, the reviewer is correct to assume that Pfeifer’s wave-line technique was abandoned for primary cleft lip repair in patients with a complete UCLP by our center. The decision to solely rely on Randall’s technique for primary cleft lip repair was based on the senior surgeons’ empirical values in conjunction with the results presented in this study.  

Whilst we appreciate that the number of cases included in this study is rather small, we strongly believe that our study has value, given the complexity and the paucity of data with regards to this specific topic. To date, and despite cleft lip repair being one of the most commonly performed procedures in cleft surgery, there is still a significant lack of data comparing outcomes related to different techniques for cleft lip repair. Furthermore, there is no consensus among centers and cleft surgeons, which technique best addresses the underlying cleft lip deformity. This comes down to a variety of reasons: 1. numerous techniques are currently used to correct the underlying cleft lip deformity; 2. studies aiming to compare outcomes in this regard often included patients with different cleft types and, furthermore, analyzed these patients at different points in time; 3. different assessment tools were used with regard to aforementioned studies. 4. A surgeon’s skills and experience may additionally influence outcomes, independent from the technique applied. All of these factors add to the variability in terms of the results stated.

However, we would like to highlight the following facts regarding this manuscript:

  • cleft type: all of the patients included presented with a complete UCLP in our study.
  • all patients received the same pre-operative treatment regimen (pre-surgical orthopedics).
  • all patients included were treated at the same center according to the same surgical protocol in terms of timing.
  • Follow-up was conducted at standardized time-points.
  • 5-year follow up was chosen as to ensure, that none of the patients had undergone corrective procedures to the lip and nose.
  • None of the patients underwent any corrective procedures (lip correction, nose repair, palatal fistula repair)
  • Similar surgical steps in terms of correcting the nasal deformity were performed (no primary rhinoplasty, but correction of the alar base and the width of the nostril)
  • None of the patients presented with fixed orthodontic appliances at the time the follow-up photographs were taken.

Reviewer 3 Report

The manuscript is quite interesting and of great clinical utility, although it is a limited study and could be better developed.

To improve this work, I recommend the following:

  1. The first paragraph doesn't add much value. I suggest an introduction about the etiology of cleft patients, through the following article: Parental Risk Factors and Child Birth Data in a Matched Year and Sex Group Cleft Population: A Case-Control Study. Int J Environ Res Public Health. 2021 Apr 27;18(9):4615. doi: 10.3390/ijerph18094615. 

  1. As the aim of the study is to compare the result of two different surgical techniques, the techniques should be better described, comparing the main differences between them, advantages and disadvantages, indications, and contraindications of each.

  1. Briefly describe the Asher 99 McDade Index and the modification used.

  1. Include the study of sample size.

  1. Clarify if differences were found in patients who had orthopaedic treatment (Hotz-plate).

  1. Point 2.2. Surgical technique includes only pre-surgical and post-surgical images and could be enriched by placing illustrative images of each of the surgical techniques.

  1. The legend of table 1 should contain the meaning of all siglas ("m", "f", "r", "l"), as it is done in table 2.

  1. In section 2.3. Evaluation of Aesthetic Outcomes, I suggest adding an illustrative image of the original scale and what is written in lines 117-119 should be placed when the scale is mentioned in line 100.

  1. The standardization of bibliographic references should be done (there are differences between 1-23 and 23-30).

Author Response

REVIEWER 3

The manuscript is quite interesting and of great clinical utility, although it is a limited study and could be better developed.

To improve this work, I recommend the following:

  1. The first paragraph doesn't add much value. I suggest an introduction about the etiology of cleft patients, through the following article: Parental Risk Factors and Child Birth Data in a Matched Year and Sex Group Cleft Population: A Case-Control Study. Int J Environ Res Public Health. 2021 Apr 27;18(9):4615. doi: 10.3390/ijerph18094615. 

 Authors’ reply:

Thank you for the suggestions made. We have adapted the introduction according to the recommendations made.

  1. As the aim of the study is to compare the result of two different surgical techniques, the techniques should be better described, comparing the main differences between them, advantages and disadvantages, indications, and contraindications of each.

 Authors’ reply:

The reviewer is right and this will definitely make the manuscript more appealing to read. As a result, we have added this section to the manuscript.

  1. Briefly describe the Asher 99 McDade Index and the modification used.

 Authors’ reply:

Done

  1. Include the study of sample size.

All non-syndromic patients with a complete UCLP that were treated at our center between 1995-2005 by means of Pfeifer’s wave line-technique and Randall’s cleft lip repair were considered. In addition, more inclusion and exclusion criteria were defined, as was mentioned in the manuscript.

Clarify if differences were found in patients who had orthopaedic treatment (Hotz-plate).

Authors’ reply:

All of the patients included underwent presurgical orthodontic treatment in terms of a Hotz-plate. Thus, this point of criticism cannot further be addressed in the present study.

  1. Point 2.2. Surgical technique includes only pre-surgical and post-surgical images and could be enriched by placing illustrative images of each of the surgical techniques.

 Authors’ reply:

We thank the reviewer for the suggestions made, however, no intraoperative were added as no adequate intraoperative photographs for Pfeifer’s technique were available. We would like to apologize for these shortcomings.

  1. The legend of table 1 should contain the meaning of all siglas ("m", "f", "r", "l"), as it is done in table 2.

 Authors’ reply:

This was corrected in the manuscript.

  1. In section 2.3. Evaluation of Aesthetic Outcomes, I suggest adding an illustrative image of the original scale and what is written in lines 117-119 should be placed when the scale is mentioned in line 100.

 Done

  1. The standardization of bibliographic references should be done (there are differences between 1-23 and 23-30

Done

Round 2

Reviewer 1 Report

This study is well writtened and described. However,  the rating system is based on the  subjective view points from the raters not on objective measurements. The Pfeifer's technique and the Randall's technique possibly can be expected there would be difference on the cleft side lip height ( measurement from Cupid's bow to the columelar base. It would be better if add some objective measurements. 

Author Response

This study is well writtened and described. However, the rating system is based on the subjective view points from the raters not on objective measurements. The Pfeifer's technique and the Randall's technique possibly can be expected there would be difference on the cleft side lip height ( measurement from Cupid's bow to the columelar base. It would be better if add some objective measurements. 

Reply to the reviewer: 

Thank you very much for the suggestions made regarding the methodology of our study. While we appreciate the potential benefit of objective measurements, this paper's primary aim was to identify potential differences regarding postoperative outcomes between two techniques for cleft lip repair based on subjective ratings. 

For this purpose, well-validated tools (Asher McDade index) and methodology (5 raters, 2 rounds of assessment; blinded etc.) were used to identify if there was a perceived difference when comparing Peifer's wave line technique with Randall's triangular flap technique in terms of the nasolabial appearance, the vermilion border and philtrum length and in the nasal frontal view. 

While no differences regarding the parameters 'nasolabial profile' and 'nasal frontal view' were determined, statistically significant differences (p=0.046) between the groups compared in terms of the parameter 'vermilion border/philtrum length' were observed. The latter parameter focuses on the philtrum length and correct alignment of the vermilion border. 

Thus, the differences detected on a subjective level suggest that the alignment of the vermilion border together with a the philtrum length were perceived as superior in group Randall in comparison to Pfeifer's approach. It is hypothesised that unilateral planning error (e.g. on the medial side of the cleft) regarding Pfeifer's approach may complicate correct alignment of the vermilion border, whereas bilateral planning error may end up with a shortening on the cleft side. With reference to objective measurements we suggest that the length of the lip can be assessed, however, that shortcomings of such objective measurements refer to detecting if the vermilion border was aligned correctly. Furthermore, we suggest that the lip length determined on the cleft side will depend on how accurately and symmetrically the Cupid's bow was designed (e.g. left-sided unilateral cleft: philtrum width right side 5mm, left side 8mm). Hence, we propose that objective measurements may add information, however, may also be linked to considerable shortcomings, as outlined in this paragraph. As a result, we decided to solely rely on subjective measurements when designing this study, because the methodology has been well-validated and well-described in the literature.